# Non-Muscle MLCK Contributes to Endothelial Cell Hyper-Proliferation through the ERK Pathway as a Mechanism for Vascular Remodeling in Pulmonary Hypertension

**DOI:** 10.3390/ijms232113641

**Published:** 2022-11-07

**Authors:** Mariam Anis, Janae Gonzales, Rachel Halstrom, Noman Baig, Cat Humpal, Regaina Demeritte, Yulia Epshtein, Jeffrey R. Jacobson, Dustin R. Fraidenburg

**Affiliations:** 1Northwestern Medical Group, Lake Forest, IL 60045, USA; 2Department of Medicine, University of Illinois at Chicago, Chicago, IL 60612, USA

**Keywords:** pulmonary hypertension, endothelial cells, myosin light chain kinase, cytoskeleton, vascular remodeling

## Abstract

Pulmonary arterial hypertension (PAH) is characterized by endothelial dysfunction, uncontrolled proliferation and migration of pulmonary arterial endothelial cells leading to increased pulmonary vascular resistance resulting in great morbidity and poor survival. Bone morphogenetic protein receptor II (BMPR2) plays an important role in the pathogenesis of PAH as the most common genetic mutation. Non-muscle myosin light chain kinase (nmMLCK) is an essential component of the cellular cytoskeleton and recent studies have shown that increased nmMLCK activity regulates biological processes in various pulmonary diseases such as asthma and acute lung injury. In this study, we aimed to discover the role of nmMLCK in the proliferation and migration of pulmonary arterial endothelial cells (HPAECs) in the pathogenesis of PAH. We used two cellular models relevant to the pathobiology of PAH including BMPR2 silenced and vascular endothelial growth factor (VEGF) stimulated HPAECs. Both models demonstrated an increase in nmMLCK activity along with a robust increase in cellular proliferation, inflammation, and cellular migration. The upregulated nmMLCK activity was also associated with increased ERK expression pointing towards a potential integral cytoplasmic interaction. Mechanistically, we confirmed that when nmMLCK is inhibited by MLCK selective inhibitor (ML-7), proliferation and migration are attenuated. In conclusion, our results demonstrate that nmMLCK upregulation in association with increased ERK expression may contribute to the pathogenesis of PAHby stimulating cellular proliferation and migration.

## 1. Introduction

Pulmonary arterial hypertension (PAH) is a severe and progressive disease characterized by obstruction of small pulmonary arteries leading to increased pulmonary vascular resistance and right heart failure [1,2]. Despite several new advancements and therapies for PAH in the recent years, mortality remains unacceptably high with a 3-year rate of 22% [2]. Genetic studies have demonstrated that mutations in bone morphogenic protein receptor type 2 (BMPR2) are present in 80% of hereditable PAH leading to loss of function and reduced downstream signaling [3,4,5,6]. Decreased BMPR2 expression is also identified in many other forms of PAH not associated with clear BMPR2 mutations [7]. However, the mechanism by which BMPR2 deficiency causes PAH is under ongoing examination with several pathways and cell types being explored [4,8,9,10,11]. In the lung, BMPR2 is highly expressed on the vascular endothelium of pulmonary arteries [9]. BMPR2 is known to interact with the cytoskeleton as it directly binds and modulates proteins related to cytoskeletal organization, including LIM domain kinase (LIMK), light chain of cytoplasmic dynein (TCTEX), and non-receptor tyrosine kinase family (SRC) [12,13,14], and has been shown to regulate cytoskeletal functions including adhesion [15] and migration [16]. BMPR2 loss has been shown to induce increased endothelial cell permeability leading to increased inflammation, in turn resulting in development of PAH [15]. Cytoskeletal defects are thus broadly seen in PAH patients and could be mechanistically linked to BMPR2 dysfunction. Endothelial cell hyper-proliferation, inhibited apoptosis, and alterations of biochemical-metabolic pathways are the unifying pathobiology of the disease [17]. Several clinical studies have provided evidence that the pulmonary endothelium is essential for the production of various mediators of vascular remodeling including vasoactive peptides, nitric oxide (NO), prostaglandin -I2 (PGI2), endothelin-1 (ET-1), fibroblast growth factor (FGF), angiotensin II (Ang II), cytokines (IL-1, IL-6), and cross talk between endothelial and smooth muscle cells; which are crucial to the development and progression of pulmonary hypertension [11,18].

BMPR2 has been a significant focus of experimental and human PAH over the past several years; however, the signaling of vascular endothelial growth factor (VEGF) and its effects on abnormal angiogenesis is also gaining increasing recognition. VEGF is abundant in the lungs with several functions including maintenance of the pulmonary endothelium [19,20]. VEGF also influences the pulmonary vasculature and aids in the production of nitric oxide (NO) and prostacyclin, which regulates vasoconstriction and dilatation, an important stimulus for PAH development and therapy [19,20]. Abnormal VEGF signaling has been identified in several disease processes including tumor angiogenesis [21] and roles in pulmonary pathologies including obstructive lung diseases [22,23] and lung injury [24]. Patients with PAH have elevated plasma levels of VEGF [25,26,27] and plexiform lesions of explanted lungs demonstrate increased levels of VEGF [28,29]. Experimental animal models chronically exposed to hypoxia express increased levels of VEGF and develop PAH [30,31,32]. Both human and experimental studies implicate a role for VEGF in the hyperproliferation of pulmonary arterial cells and hypoxia-induced vascular remodeling; however, the extensive physiologic involvement of VEGF on endothelial cells makes exploration of specific cell signaling in the pathogenesis of PAH challenging. At the molecular level, the proangiogenic effects of VEGF include cellular proliferation, migration, and reorganization of the actin cytoskeleton and some effects have been associated with MAPK/ERK signaling [33,34]. Similarly, the pro-proliferative and apoptosis resistant phenotype of pulmonary artery endothelial cells (HPAEC) in the absence of BMPR2 has been linked to activation of ERK1/2 and p38MAPK [35], creating overlap in the signaling of two important stimuli in the pathobiology of PAH.

The role of endothelial cells has been increasingly recognized in the pathogenesis of PAH, but there is still much left to be explored. Additionally, although increased expression of myosin light chain kinase (MLCK) in pulmonary arterial smooth muscle cells (PASMCs) from patients with PAH compared with controls has been established [36], the same does not hold true for HPAEC. MLCK, a central cytoskeletal regulator is a Ca^2+^/Calmodulin dependent enzyme encoded by myosin light chain kinase encoding gene (MYLK)that phosphorylates myosin light chain (MLC) and plays a key pathophysiological role in complex diseases including acute lung injury (ALI) and asthma [37]. Garcia et al. first identified a non-muscle MLCK (nmMLCK) that encodes four high molecular weight MLCK isoforms (MLCK1-4) in the pulmonary endothelium [38]. A major distinguishing feature of the two isoforms is that nmMLCK contains an additional 922-amino-acid stretch at the N-terminus that is not present in smooth muscle MLCK (smMLCK). This amino-acid stretch is involved in distinct cellular functions through unique interactions with other contractile proteins and is shown to influence junctional disruption and paracellular gap formation, cell division, proliferation and cell shape [39].

Therefore, the aim of this study is to examine the downstream effects of nmMLCK activation on cellular proliferation and migration in HPAECs in the presence of two pathologically relevant stimuli in the pathogenesis of PAH, namely BMPR2 deficiency and VEGF stimulation. Utilizing cell proliferation and migration, we hypothesize that increased nmMLCK activity leads to increased ERK/MAPK activity, and further suggesting that nmMLCK activity is important in the development and progression of PAH.

## 2. Results

### 2.1. Upregulated Protein Expression of nmMLCK and ERK in BMPR2 Silenced Human Pulmonary Artery Endothelial Cells

BMPR2 mutations and the downregulation of BMPR2 expression is known to contribute to the pathogenesis of PAH. As BMPR2 is highly expressed in the endothelium of pulmonary arteries, we chose to conduct our experiments on normal HPAECs. To examine a potential mechanistic relationship between BMPR2 and the cytoskeleton, we silenced BMPR2 in HPAECs and used Western blot analysis to measure protein levels of cytoskeletal regulating pathways. Myosin light chain when phosphorylated leads to activation of the quiescent endothelium and cytoskeletal reorganization defined by stress fiber formation and contractility. ERK is a known upstream regulator of MLCK and phosphorylated-ERK (p-ERK) is the activated form. After 48 h of targeted knockdown, BMPR2 protein levels were reduced by >90% (Figure 1A,B; *p* ≤ 0.01). BMPR2 silenced HPAECs demonstrated an approximately 10-fold increase in phosphorylated-myosin light chain (p-MLC) protein expression when compared to control (Figure 1A,C; *p* ≤ 0.01). The upregulated expression of nmMLCK was associated with a nearly five-fold increase in expression of p-ERK (Figure 1A,D; *p* ≤ 0.01). These results indicate that repression of BMPR2 in HPAECs leads to activation of both nmMLCK and ERK pathways, which are known to play an important role in cytoskeletal regulation.

### 2.2. BMPR2 Silencing Induces Increased Endothelial Cell Viability, Proliferation, and Cytokine Release Which Are Attenuated by MLCK Inhibition

Since BMPR2 silencing demonstrated upregulation of cytoskeletal proteins, we next sought to evaluate for the endothelial dysfunction that is characteristic of vascular remodeling. Endothelial dysfunction in PAH leads to a hyperproliferative and apoptosis-resistant phenotype; to evaluate for this, BMPR2 silenced HPAECs were subjected to established measurements of cellular proliferation including Western blot analysis of proliferating cell nuclear antigen (PCNA) protein expression and water-soluble tetrazolium salt-1 (WST-1) assay. PCNA is a component of cell replication machinery and WST-1 measures the activity of cellular mitochondrial dehydrogenases. BMPR2 silenced HPAECs demonstrated over 200% increase in PCNA protein expression (Figure 2A,B; *p* ≤ 0.05) and approximately 50% increase in endothelial cell proliferation measured by WST-1 assay when compared to control (Figure 2C; *p* ≤ 0.01). To evaluate the potential role of the cytoskeleton in the hyperproliferation of BMPR2 silenced HPAECs, an MLCK specific inhibitor was used to prevent phosphorylation of MLC. Pre-treatment with ML-7 in conjunction with BMPR2 silencing led to a 50% decrease in cell viability and proliferation when compared to BMPR2 silencing alone (Figure 2C; *p* ≤ 0.01).

Endothelial dysfunction was also evaluated in the form of inflammation by measurement of cytokine release. Interleukins 6 and 8 are known to be upregulated in PAH and were measured in our BMPR2 silenced HPAECs by ELISA technique. In Figure 2D,E, IL-6 (*p* ≤ 0.05) and IL-8 (*p* ≤ 0.001) secretion were both increased in the basal media of BMPR2 silenced HPAECs when compared to control. MLCK specific inhibition with ML-7 pre-treatment prevented the increased cytokine release induced by BMPR2 known down (IL-6 *p* < 0.05; IL-8 *p* ≤ 0.001). Taken together, BMPR2 silencing in HPAECs leads to endothelial dysfunction as measured by increased cellular proliferation, viability, and cytokine release, which are all attenuated by inhibition of the nmMLCK pathway.

### 2.3. BMPR2 Silencing Significantly Increases nmMLCK Dependant HPAEC Migration

Endothelial dysfunction in PAH is also represented by increased and disorganized cellular migration. We next sought to evaluate migration in our BMPR2 silenced HPAECs and how cellular motility is affected by MLCK inhibition. To measure migration, the cell monolayer is disrupted, and the rate of recovery is measured; electrical cell impedance sensing (ECIS) wound assay and scratch assay are two established modalities for measuring isolated cell migration. BMPR2 transfected HPAECs were grown to confluence on electrodes, subjected to ECIS-based wounding, and transendothelial resistance (TER) was measured over time, representing the rate of recovery. In Figure 3A, BMPR2 silenced HPAECs demonstrate a trend toward increased rate of recovery after wounding and increased area under the curve when compared to control. Pre-treatment with MLCK inhibitor in conjunction with BMPR2 silencing reduces HPAEC migration (*p* ≤ 0.01). Cellular migration was also evaluated by wound healing scratch assay. By this method, a scratch is created in the monolayer of HPAECs with respective treatments, the size of the created wound is measured over time, and the rate of wound closure is representative of cellular migration. BMPR2 transfected HPAECs demonstrated a nonsignificant trend towards increased wound closure at 24 h (Figure 3C,D). Pre-treatment with ML-7 in BMPR2 transfected HPAECs had a significant reduction in endothelial cell motility and migration when compared to BMPR2 transfected HPAECs alone (Figure 3C,D; *p* < 0.05). Taken together, MLCK inhibition has a significant effect on migration in BMPR2 deficient HPAECs.

### 2.4. VEGF Treatment Leads to Increased nmMLCK Activity

BMPR2 deficiency in HPAECs led to endothelial dysfunction seen in PAH characterized by hyperproliferation and cytokine release; these processes were dependent on the nmMLCK pathway, as specific inhibition negated these adverse effects. To further validate the importance of nmMLCK activation on the early cellular mechanisms that lead to the development of PAH, VEGF stimulation was also explored in a similar context. HPAECs were treated with VEGF for 72 h and demonstrated increased expression of p-MLC when compared to vehicle control, consistent with increased nmMLCK activity (Figure 4A,B; *p* ≤ 0.01). Pre-treatment with MLCK specific inhibitor, ML-7 on VEGF-treated HPAECs decreased expression of p-MLC (Figure 4A,B; *p* ≤ 0.001). Similarly, VEGF treated HPAECs demonstrated upregulated p-ERK (Figure 4A,C) with decreased expression after ML-7 pretreatment (Figure 4A,C; *p* ≤ 0.001).

### 2.5. VEGF Treatment Increases Endothelial Cell Proliferation and Cytokine Release Which Is Negated by MLCK Inhibition

Endothelial dysfunction measured by hyperproliferation and inflammation was also measured in VEGF stimulated HPAECs. After 72 h, VEGF-treated HPAECs demonstrated a non-significant trend toward increased cellular proliferation compared to vehicle control measured by PCNA expression (Figure 5A,B). Significantly increased cellular viability and proliferation was measured by WST assay in VEGF-treated HPAECs compared to control (Figure 5C; *p* ≤ 0.001). ML-7 pre-treatment reduced hyperproliferation and viability in VEGF-treated cells when compared to VEGF treatment alone both by PCNA expression (Figure 5A,B; *p* ≤ 0.01) and cellular proliferation (WST-1) assay (Figure 5C; *p* ≤ 0.001). VEGF stimulation leads to a hyperproliferative phenotype in HPAECs which is mediated by the nmMLCK pathway.

### 2.6. VEGF Treatment Significantly Increases nmMLCK Dependant HPAEC Migration 

Lastly, VEGF-treated cells were also measured for hypermigration measured by both ECIS-based wounding and traditional scratch assay. VEGF treatment enhanced HPAEC migration compared to control as shown in Figure 6A, and pre-treatment with ML-7 abrogated this increased migration. Area under the curve analysis showed a significant increase in cellular motility in VEGF treated HPAECs compared to control (Figure 6B; *p* ≤ 0.01), and pre-treatment with ML-7 decreased the HPAEC migration noted in VEGF treatment alone (*p* < 0.05). These findings were confirmed with scratch assay, which demonstrated increased wound gap closure after 24 h in VEGF treatment compared to control (Figure 6C,D; *p* < 0.05), and similar attenuation of migration with MLCK specific inhibition (Figure 6C,D; *p* ≤ 0.001). Overall, these results confirm that in addition to BMPR2 silencing, HPAECs treated with VEGF, another known PAH-inducing cellular mechanism, leads to endothelial dysfunction and hyper-migration mediated by the nmMLCK pathway.

## 3. Discussion

Human pulmonary endothelial cells from patients with idiopathic pulmonary arterial hypertension are known to grow faster in culture due to both increased proliferation and resistance to apoptosis [40]. These cells also demonstrate increased cellular migration due to endothelial dysfunction [41]. BMPR2 silencing in pulmonary endothelial cells has reiterated the cellular dysfunction noted in PAH pathogenesis [10,42,43,44,45,46]. BMP/BMPR2 effects mediate SMAD 1/5/8 phosphorylation leading to characteristic cellular phenotype observed in PAH [47]. There are SMAD independent pathways such as Wnt/catenin and PPAR which also regulate cell signaling [48,49].

Pulmonary arterial endothelial cell proliferation and migration play a pivotal role in PAH pathogenesis. In this study, we identified that nmMLCK upregulation in human pulmonary endothelial cells is associated with many of the abnormalities and pathogenic mechanisms observed in PAH. The effect of nmMLCK in driving PAH pathogenesis is further strengthened by the loss of this effect when MLCK is inhibited which we have also shown to have effect on the ERK pathway. Human PAEC transfected with BMPR2 siRNA or treated with VEGF showed increased proliferation both by WST assay and protein expression when compared to scrambled siRNA or control conditions. Increased migration was observed after VEGF treatment in both cell scratch and ECIS wound healing assays. Treatment of BMPR2 silenced and VEGF treated HPAECs with the MLCK specific inhibitor (ML-7) resulted in decreased proliferation and migration further promoting the novel identification of nmMLCK playing an important role in PAH pathogenesis. The upregulated nmMLCK expression was associated with increased ERK production in endothelial cells as a likely mechanism of increased proliferation and migration under BMPR2 deficiency and VEGF stimulation. This work demonstrates that non-muscle MLCK is likely an important contributor to endothelial dysfunction recognized in PAH and may represent a unique therapeutic target (Figure 7).

The human MYLK gene spanning 217.6 kb on chromosome 3q21.1 encodes three isoforms including non-muscle MLCK isoform (nmMLCK), smooth muscle isoform (smMLCK) and telokin (KRP), a small myosin filament-binding protein [50]. Vascular endothelial cells, only express the non-muscle MLCK isoform, which contains a novel NH2-terminus stretch (amino acid 1–922) not present in the open reading frame of smooth muscle MLCK [39,51]. Furthermore, the chromosome location of MYLK (3q21) is an active site for several inflammatory disorders including asthma, allergic rhinitis, COPD and atopic dermatitis. Both smMLCK and nmMLCK phosphorylate myosin light chains to regulate cellular contraction and relaxation along with barrier function in turn playing an important role in the pathogenesis of various disease processes including asthma and acute lung injury [38,52,53]. Since the identification of non-muscle MLCK, various studies have elucidated that nmMLCK is unique in structure [38].

We explored the significance of BMPR2 signaling involvement in cytoskeletal structure and function and its link to nmMLCK. Non muscle isoform of MLCK has itself been described as vital in the rapid dynamic coordination of the cytoskeleton involved in cancer cell proliferation and migration in ways similar to the tumor like growth of pulmonary endothelial cells in PAH [54]. Previous literature has demonstrated that both patients with PAH and endothelial cell models have increased levels of ERK [55]. Recently, Awad et al. show that Raf family members and ERK1/2 are activated after BMPR2 knockdown [35]. We were able to show that BMPR2 silencing in HPAECs is linked to increased expression of nmMLCK along with an increase ERK phosphorhylation indicative of a potential association between ERK, BMPR2 and nmMLCK. In previous studies, VEGF stimulation has led to increased angiogenesis in endothelial cells through various mechanisms [56,57,58]. It is well recognized that VEGF expression is elevated in arterial cells of the characteristic plexogenic lesions of patients with advanced pulmonary hypertension [59]. Similarly, cell proliferation and migration were both ascertained to be significantly increased in our in vitro model when VEGF was employed as an angiogenic stimulus. Our data provides insight into cellular migration enhanced as a function of cytoskeletal reorganization mediated by nmMLCK activation in the presence of VEGF stimulation in HPAECs.

ML-7, a selective inhibitor of MLCK, acts on the adenosine triphosphate (ATP)-binding site of the active center of MLCK [60,61,62]. We were able to validate our results with the decreased activity of nmMLCK by showing reduced expression of p-MLC with the use of ML-7 along with a concurrent decrease in ERK/MAPK pathway. Hence, establishing the link between nmMLCK and ERK in the various processes of cell proliferation and migration which are the cornerstone of PAH pathogenesis. In the future, exploring the transcriptional link between the two and further corroborating findings with additional nmMLCK inhibitors will be beneficial. Previous work from our group with use of MLCK specific inhibition has also implicated the cytoskeleton and prevention of MLC phosphorylation in hemin-induced endothelial dysfunction in the context of PH due to chronic hemolysis [60]. These two studies, in conjunction, support the need for further exploration of MLCK inhibition as a potential therapy for PAH.

This work is limited to the use of cell models that recapitulate the endothelial dysfunction that is seen during vascular remodeling in the pathogenesis of PAH. We utilized cell models that we think have strong relevance to human disease, particularly with reduction in BMPR2 levels as seen in numerous PAH patient subgroups as well as stimulation of the cells with a growth factor, VEGF, that is known to be elevated in patients with some forms of PAH [15,25,26,27]. Further study will require the use of experimental animal models of pulmonary hypertension as well as cells and tissues from human PAH in order to better understand the role of nmMLCK in endothelial dysfunction as well as development and progression of human PAH.

In summary, the data from this study is the first in our knowledge to recognize the importance of increased expression of nmMLCK contributing to endothelial cellular proliferation and migration with downstream activation of Ras/Raf/ERK pathway. The enhanced nmMLCK activity appears to play a crucial role in the pathobiology of PAH. The nmMLCK-Raf/ERK link presents a novel pathway for development of more efficient potential targets for treatment of pulmonary hypertension.

## 4. Materials and Methods

### 4.1. Cell Culture

Normal HPAECs were cultured at passages 5–8 in endothelial basal medium-2 (EBM-2) supplemented with growth factors [endothelial growth medium (EGM)-2 Single Quot kit from Lonza (Basel, Switzerland)] and containing 10% fetal bovine serum (FBS). Cells were maintained at 37 °C in a humidified incubator with 5% CO_2_ and 95% air. Primary HPAECs were seeded at a density of 180,000–200,000 cells/well in 6-well plates for RNA and protein analysis, respectively. For all experiments, basal media was replaced with EGM-2 containing 2% FBS with added treatment or control conditions as described in each individual experiment.

### 4.2. BMPR2 Transfection

HPAECs were transfected with gene-specific siRNA pools targeting BMPR2 non-specific siRNA (Ambion, Austin, TX, USA) at a final concentration based on the culture vessel surface area per XFECT-1 (Clontech, Mountain View, CA, USA) protocol for 4 h followed by growth in EGM-2 containing 2% charcoal-stripped serum. Nontargeting siRNA pool-1 (siGENOME, Dharmacon, Lafayette, CO, USA) was used as a control. After incubating for an additional 46 h (48 h from the start of transfection), total RNA or protein lysates were collected for Western blot analysis. BMPR2 silencing was repeated with addition of nmMLCK inhibitor, ML-7 hydrochloride [1-(5-iodonaphthalene-1-sulfonyl)-[1H]-hexahydro-1,4-diazepine hydrochloride; (Tocris Bioscience, Bristol, UK) at 10 μM at 4 h time point followed by incubation for 42 h.

### 4.3. Western Blotting

Whole cell protein lysates were isolated from HPAEC with RIPA buffer (Millipore, Burlington, MA, USA) supplemented with protease and phosphatase inhibitor cocktail (Thermo Fisher Scientific, Waltham, MA, USA). Protein lysates were resolved by SDS-PAGE and transferred to a nitrocellulose membrane with Bio-Rad Laboratories (Hercules, CA, USA) Western blotting system. Membranes were incubated with 5% blocking solution (nonfat milk) in PBS-Tween 20 (0.1%) for 1 h at room temperature and then incubated overnight at 4 °C with primary antibody. The following day, membranes were incubated in secondary antibodies and then visualized with chemiluminescence (Thermo Fisher Scientific). Antibodies against p-MLC (1:1000), p-ERK (1:1000), total ERK (1:1000), and proliferating cell nuclear antigen (PCNA) (1:500) were purchased from Proteintech Group Inc (Rosemont, IL, USA). Anti-BMPR2 antibody (1:500) was obtained from Abcam (Cambridge, MA, USA). Quantitative data was obtained using Image J (Version 1.46r, National Institutes of Health, Bethesda, MA, USA) and data is presented as mean relative protein expression ± standard error (SE). *n* is defined as protein lysates extracted from a single well for a given condition.

### 4.4. Cell Proliferation and Viability Assay

HPAECs were plated at 0.1–5 × 10^4^/well in a 96-well microplate with a final volume of 100 μL/well. Cell proliferation and viability was assessed after treatment with addition of 10 μL/well WST-1/ECS (Millipore Sigma) solution to each well for 4 h. Cell proliferation in this assay is based on cleavage of the tetrazolium salt WST-1 to formazan by cellular mitochondrial dehydrogenases and an increase in formazan dye. Cell proliferation was quantified by a multi well spectrophotometer (microplate reader) by measuring the absorbance of dye solution at 450 nm and reference wavelength 650 nm. Quantitative data is presented as mean ± SE. *n* is defined a single well for a given condition.

### 4.5. Enzyme Linked Immunosorbent Assay (ELISA)

HPAECs were grown on 6-well plates and treated with respective conditions once confluent. The cell culture media was removed and centrifuged and the supernatant was used for ELISA analysis. Commercially available sandwich ELISA kits were purchased from BioLegend (San Diego, CA, USA) and measurement of IL-6 and IL-8 were obtained according to manufacturers instructions. A microplate reader was used to measure corresponding absorbance. Data is presented as mean ± SE. *n* is defined a single well for a given condition.

### 4.6. Electric Cell-Substrate Impedance Sensing (ECIS) Wound Assay

ECIS system (Applied Biophysics, Inc., USA) and 8W1E well arrays were used to detect and track HPAECs migration during electrical wound-healing assay. HPAECs were seeded at a density of 200,000–250,000 cells/well and treated with respective conditions 24 h after attachment. ECIS plate was then transferred to the humidified 5% CO_2_ incubator at 37 °C and wells were allowed to equilibrate in the incubator. Wound was then applied at 40,000 Hz and 3 mA for 10 s per well. Over the next 24 h, the resistance from each well was measured every 15 min and then analyzed. *n* is defined as each independent well for a given condition. Area under the curve (AUC) was calculated for each condition and presented as mean ± SE.

### 4.7. Scratch Assay

Cell motility was assessed with cell scratch assay [61]. The cell monolayer was scratched with a sterile P20 pipette tip and the debris was removed by washing twice with warm working media, and then replaced with fresh working medium after which images were captured at 0 and 24 h with inverted microscope with a digital camera (Nikon Eclipse TE2000-s, Nikon Instruments Inc., Melville, NY, USA) at 4× *g* or 10× *g* magnification [62]. Quantification for gap closure was completed using Image J software, as described previously [63] and data is presented as mean ± SE. *n* is defined a single well for a given condition.

### 4.8. Statistical Analysis

SigmaPlot software (v14, Systat Software Inc., Palo Alto, CA, USA) was used for Student’s *t*-test to calculate significance between two groups and One-Way ANOVA for many groups. Composite data are shown as the mean ± standard error. A *p* value of ≤ 0.05 was considered significant.

## Figures and Tables

**Figure 1 ijms-23-13641-f001:**
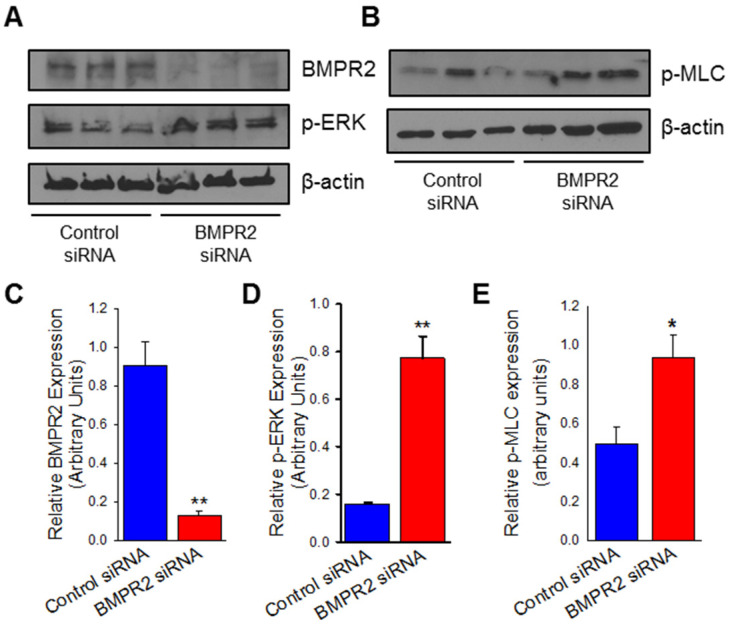
BMPR2 silencing in human pulmonary artery endothelial cells (HPAECs) is associated with ERK/MAPK activation. Representative Western blot images denoting BMPR2 and p-ERK (**A**) as well as p-MLC (**B**) protein expression between HPAECs transfected with BMPR2 siRNA or control siRNA for 48 h. Bar graphs summarizing relative BMPR2 (**C**), phosphorylated ERK (**D**), and phosphorylated MLC (**E**) protein expression at 48 h in BMPR2 silenced HPAECs compared to control (*n* = 3). * indicates *p* < 0.05; **, *p* ≤ 0.01; BMPR2, Bone morphogenetic protein receptor type II; p-ERK, phosphorylated extracellular signal-regulated kinase; p-MLC, phosphorylated myosin light-chain; siRNA, small interfering (silencing) RNA.

**Figure 2 ijms-23-13641-f002:**
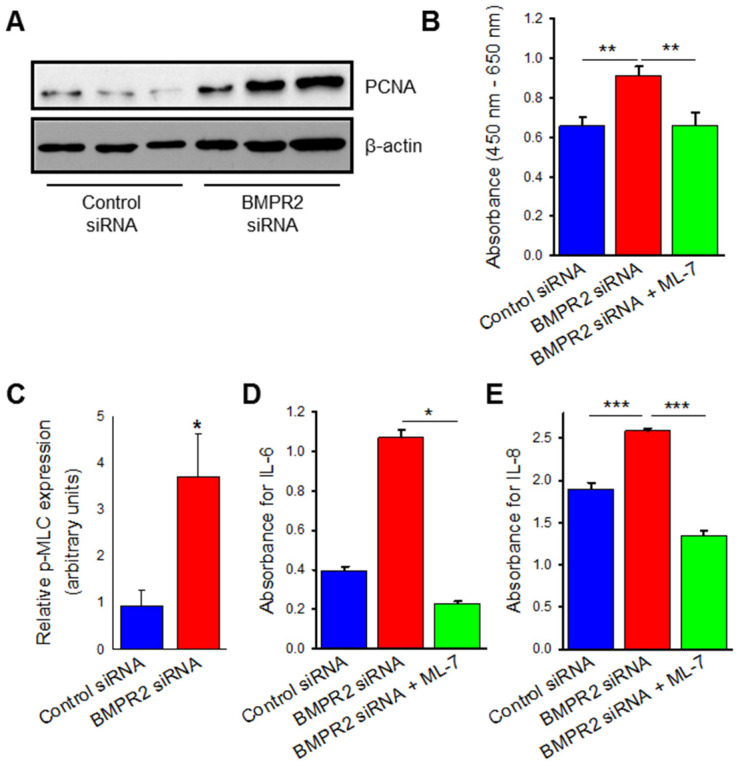
BMPR2 silencing increases HPAEC proliferation and cytokine release which is attenuated by inhibition of myosin light chain kinase (MLCK). Representative Western blot (**A**) and accompanying bar graph (**B**) depicting PCNA protein expression at 48 h between HPAECs transfected with BMPR2 or control siRNA (*n* = 3). (**C**) Bar graph representing changes in proliferation and viability as measured by WST-1 assay in HPAECs transfected with BMPR2 siRNA, control siRNA, and BMPR2 siRNA with ML-7 (10 μM) pre-treatment for 48 h (*n* = 9). Bar graph measuring IL-6 (**D**) and IL-8 (**E**) concentrations in the media of HPAECs transfected with BMPR2 siRNA, control siRNA, and BMPR2 siRNA with ML-7 (10 μM) pre-treatment for 48 h (*n* = 3). * indicates *p* < 0.05; **, *p* ≤ 0.01; ***, *p* ≤ 0.001; PCNA, proliferating cell nuclear antigen; BMPR2, Bone morphogenetic protein receptor type II; siRNA, small interfering (silencing) RNA; ML-7, myosin light chain kinase specific inhibitor; IL-6, Interleukin-6; IL-8, Interleukin-8.

**Figure 3 ijms-23-13641-f003:**
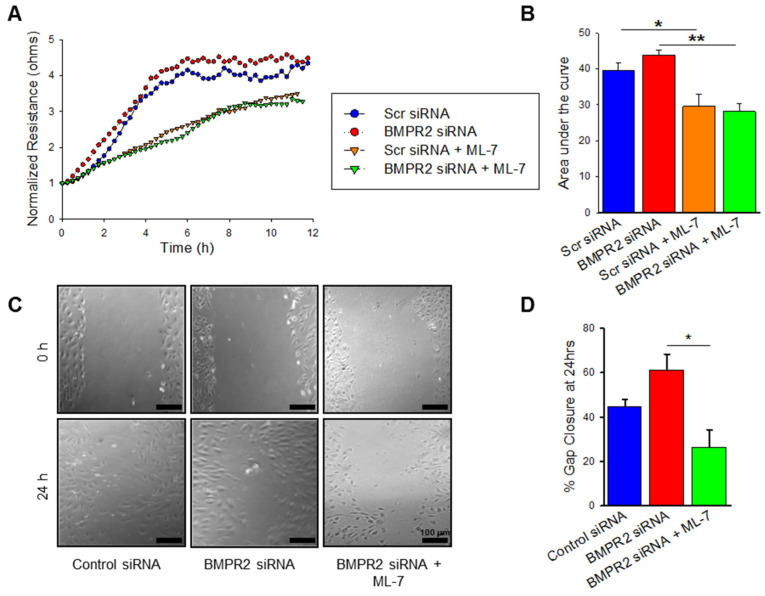
BMPR2 silencing increases HPAEC migration which is attenuated by MLCK inhibition. (**A**) Plot demonstrating the transendothelial resistance by ECIS-based wounding over time in HPAECs transfected with BMPR2 siRNA, control siRNA, BMPR2 siRNA with ML-7 (10 μM), and control siRNA with ML-7 (10 μM) (*n* = 2). (**B**) Bar graph denoting the area under the curve measurements at 12 h for the ECIS-based wounding experiments (*n* = 2). (**C**) Representative images of wound healing assay depicting scratches created in confluent cultures of HPAECs transfected with BMPR2 siRNA, control siRNA, and BMPR2 siRNA with ML-7 (10 μM) at 0 h and 24 h time points; scale bar = 100 μm. (**D**) Bar graph summarizing percent gap closure at 24 h for the wound healing assay experiments (*n* = 3). * indicates *p* < 0.05; **, *p* ≤ 0.01; BMPR2, Bone morphogenetic protein receptor type II; siRNA, small interfering (silencing) RNA; ML-7, myosin light chain kinase specific inhibitor.

**Figure 4 ijms-23-13641-f004:**
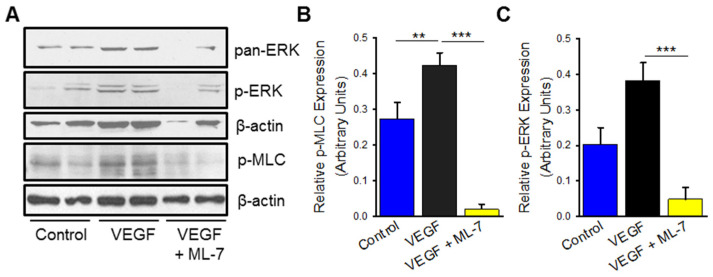
VEGF stimulation in HPAECs leads to upregulation of ERK/MAPK, which is abrogated by MLCK inhibition. (**A**) Representative Western blot images denoting p-ERK and p-MLC protein expression in HPAECs treated with VEGF (100 ng/mL), control (PBS vehicle), and VEGF with ML-7 (10 μM) pre-treatment for 72 h. Bar graphs summarizing relative phosphorylated ERK (**B**) and phosphorylated MLC (**C**) protein expression at 72 h in HPAECs treated with VEGF, control, and VEGF with ML-7 (*n* = at least 4 for p-ERK, *n* = at least 7 for p-MLC). ** indicates *p* ≤ 0.01; ***, *p* ≤ 0.001; VEGF, vascular endothelial growth factor; p-ERK, phosphorylated extracellular signal-regulated kinase; p-MLC, phosphorylated myosin light-chain; ML-7, myosin light chain kinase specific inhibitor.

**Figure 5 ijms-23-13641-f005:**
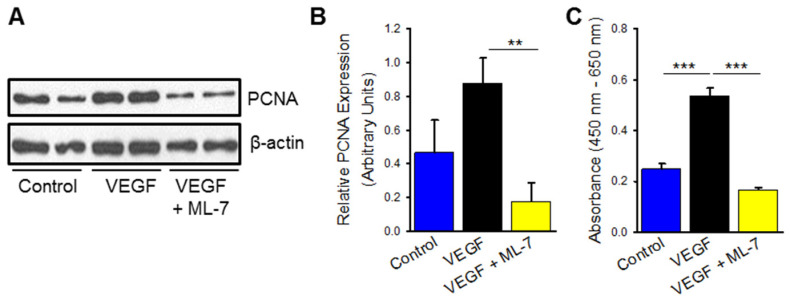
VEGF treatment increases HPAEC proliferation and cytokine release which is attenuated by inhibition of myosin light chain kinase (MLCK). Representative Western blot (**A**) and accompanying bar graph (**B**) depicting PCNA protein expression at 72 h between HPAECs treated with VEGF (100 ng/mL), control (PBS vehicle), and VEGF with ML-7 (10 μM) pre-treatment (*n* = at least 2). β-actin loading control is identical to Figure 4A as the representative blot was derived from the same experiment. (**C**) Bar graph denoting changes in proliferation and viability as measured by WST-1 assay in HPAECs treated with VEGF, control, and VEGF with ML-7 for 72 h (*n* = 4). ** indicated *p* ≤ 0.01; *** indicates *p* ≤ 0.001; PCNA, proliferating cell nuclear antigen; VEGF, vascular endothelial growth factor; ML-7, myosin light chain kinase specific inhibitor; IL-6, Interleukin-6; IL-8, Interleukin-8.

**Figure 6 ijms-23-13641-f006:**
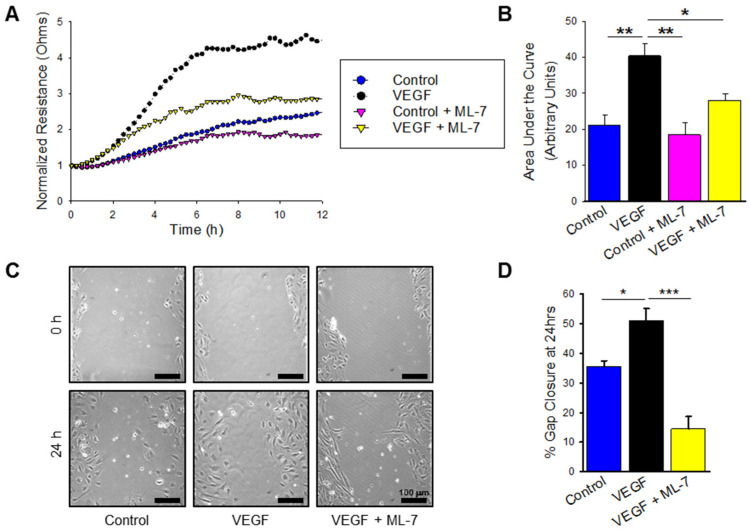
VEGF treatment increases HPAEC migration which is attenuated by MLCK inhibition. (**A**) Plot demonstrating the transendothelial resistance by ECIS-based wounding over time in HPAECs treated with VEGF (100 ng/mL), control (PBS vehicle), VEGF with ML-7 (10 μM) pre-treatment, and control with ML-7 (*n* = at least 3). (**B**) Bar graph denoting the area under the curve measurements at 12 h for the ECIS-based wounding experiments (*n* = at least 3). (**C**) Representative images of wound healing assays depicting scratches created in confluent cultures of HPAECs treated with VEGF (100 ng/mL), control (PBS vehicle), VEGF with ML-7 (10 μM) pre-treatment at 0 h and 24 h time points; scale bar = 100 μm. (**D**) Bar graph summarizing percent gap closure at 24 h for the wound healing assay experiments (*n* = 2). * indicates *p* ≤ 0.05; **, *p* ≤ 0.01; ***, *p* ≤ 0.001 VEGF, vascular endothelial growth factor; ML-7, myosin light chain kinase specific inhibitor.

**Figure 7 ijms-23-13641-f007:**
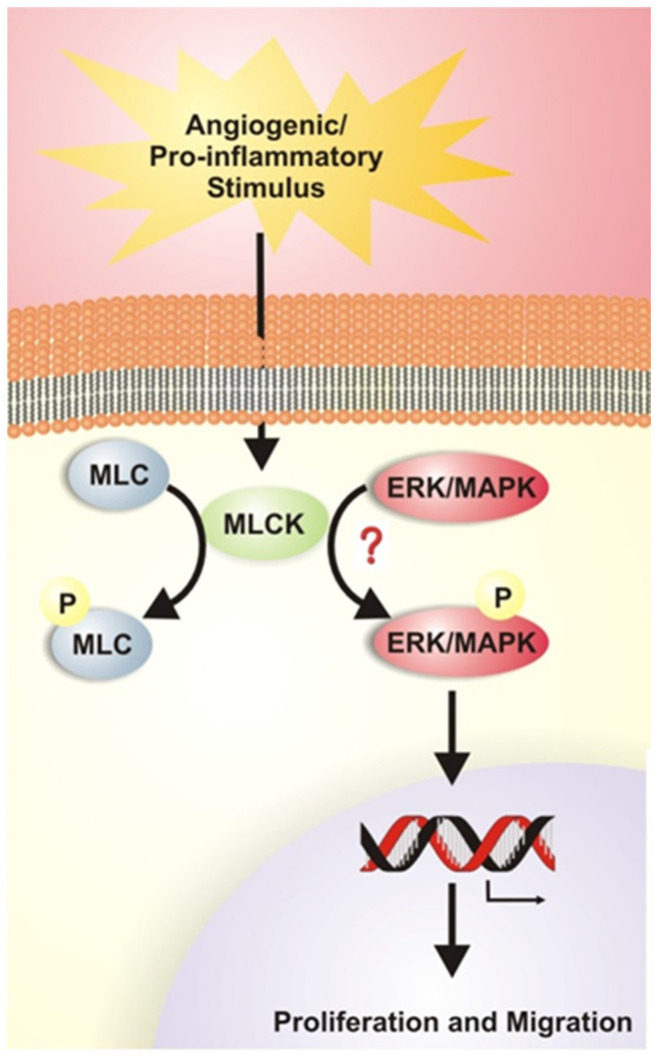
Schematic representation of our findings demonstrating that nmMLCK activation due to BMPR2 downregulation or VEGF stimulation is associated with increased ERK phosphorylation, potentially directly or indirectly, contributing to the pathogenesis of PAH by stimulating cellular proliferation and migration.

## Data Availability

Not applicable.

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
