# Peer review of "Non-Muscle MLCK Contributes to Endothelial Cell Hyper-Proliferation through the ERK Pathway as a Mechanism for Vascular Remodeling in Pulmonary Hypertension"

_ijms, 2022, doi:10.3390/ijms232113641_

Round 1

Reviewer 1 Report

I am interested in the study entitled Non-Muscle MLCK Contributes to Endothelial Cell Hyper- 2 proliferation through the ERK pathway as a Mechanism for 3 Vascular Remodeling in Pulmonary Hypertension. By Mariam Anis et al. 

Authors showed the role of non-muscle MLCK in the HPAECs dysfunction mimic pulmonary arterial hypertension. Although it may be better if there was the data as to the expression of nmMLCK in the pulmonary endothelial cells derived from PAH patients, or the analysis using PAH animal model, I think the data were analyzed properly. 

Author Response

We would like to thank the reviewers for taking the time to read and review our manuscript, “Non-Muscle MLCK Contributes to Endothelial Cell Hyper-proliferation through the ERK pathway as a Mechanism for Vascular Remodeling in Pulmonary Hypertension.” We earnestly appreciate both the reviewers for their interest on the study and their helpful, constructive comments on the manuscript. We hope the explanation and revision provide a satisfactory response to the reviewers’ concerns and questions. Comments from each reviewer are listed below with the reviewers’ unedited comments in italic text, followed by our response.

Reviewer #1

I am interested in the study entitled “Non-Muscle MLCK Contributes to Endothelial Cell Hyper- 2 proliferation through the ERK pathway as a Mechanism for 3 Vascular Remodeling in Pulmonary Hypertension.” By Mariam Anis et al. 

Authors showed the role of non-muscle MLCK in the HPAECs dysfunction mimic pulmonary arterial hypertension. Although it may be better if there was the data as to the expression of nmMLCK in the pulmonary endothelial cells derived from PAH patients, or the analysis using PAH animal model, I think the data were analyzed properly

Thank you for reviewing our manuscript and your interest in our work. We agree that having cell lines derived from PAH patients or animal models would provide additional data to strengthen our findings, unfortunately these diseased human cell lines and isolation of endothelial cells from small rodent models is not available to us at this time. We utilized cell models that we think have strong relevance to human disease, particularly with reduction in BMPR2 levels as seen in numerous PAH patient subgroups as well as stimulation of the cells with a growth factor, VEGF, that is known to be elevated in patients with some forms of PAH. We have added additional information on these limitations for the project.

Reviewer 2 Report

I have read the manuscript of Mariam Anis et al. with title Non-Muscle MLCK Contributes to Endothelial Cell Hyper- proliferation through the ERK pathway as a Mechanism for Vascular Remodeling in Pulmonary Hypertension, in where we are confirmed, that when nmMLCK is inhibited by MLCK selective inhibitor (ML-7), proliferation and migration are attenuated. Before accepting the article for publication, I have some observations that could help improve this manuscript.

Page 1 lines 26 in the abstract please replace pulmonary arterial hypertension by the abbreviation “PAH”

Page 2 line 44 please development the abbreviations LIMK, TCTEX and SRC

Page 2 line 46 could said BMPR2 loss instead of loss if BMPR2

Page 2 line 47 please replaces development of pulmonary hypertension by the abbreviation PAH

Page 2 lines 55 please replace all of which is crucial to the development ….. by which are because “all it is plural not singular

Page 2 line 62 add an abbreviation for nitric oxide

Page 2 lines 63, 69 and 72, what is PH?

Page 2 line 84 what is MYLK?

Page 2 lines 93 please add therefore the aim of this study was…..

Page 3, 4, 5, and 7 please add in each figure the corresponding abbreviations

I find it difficult to believe that the p of the significant differences is so exact throughout the entire manuscript, when one performs a statistical test, the p varies for this reason I think it is better to put them as p≤….. Please substitute this in all manuscript

I reviewed each of the westerns that were provided to me, but I never saw the p-MLC one in figure 1 and 4, please provide them, and the one provided in figure 4 does not correspond

Page 8 lines 292-299, please remove this paragraph and place it before starting the discussion because it can serve as an introduction to the discussion and replaces in same paragraph pulmonary arterial hypertension by PAH

Page 8, line 283 please substitute EC by VEC

Page 11, line 409 what sigma plot version is, please add

Page 5 figure 3 (A) please delete this graphic and leave the b because it is not necessary, since it shows the same data

The load control (β-actin) in figure 4 and 5 are the same please substitute another load control in any of the figures

it is difficult to demonstrate so much significant value in figure 2 B with respect to the western of figure 2 A, because there is more load control (β-actin) in the lane that corresponds to the PCNA and there is more expression there, when making the quotient I think the result would not be as impressive, could you please review this result.

Please add a figure that summarize your manuscript for mayor comprehension

I could add a part of the limitations of the study because a study of vascular reactivity in a chamber for an isolated organ would be great and would support the results obtained in cultured cells, I know in advance that this is difficult due to obtaining pulmonary arteries, but yes it was possible perhaps in postmortem patients

Author Response

We would like to thank the reviewers for taking the time to read and review our manuscript, “Non-Muscle MLCK Contributes to Endothelial Cell Hyper-proliferation through the ERK pathway as a Mechanism for Vascular Remodeling in Pulmonary Hypertension.” We earnestly appreciate both the reviewers for their interest on the study and their helpful, constructive comments on the manuscript. We hope the explanation and revision provide a satisfactory response to the reviewers’ concerns and questions. Comments from each reviewer are listed below with the reviewers’ unedited comments in italic text, followed by our response.

I have read the manuscript of Mariam Anis et al. with title Non-Muscle MLCK Contributes to Endothelial Cell Hyper- proliferation through the ERK pathway as a Mechanism for Vascular Remodeling in Pulmonary Hypertension, in where we are confirmed, that when nmMLCK is inhibited by MLCK selective inhibitor (ML-7), proliferation and migration are attenuated. Before accepting the article for publication, I have some observations that could help improve this manuscript.

1.Page 1 lines 26 in the abstract please replace pulmonary arterial hypertension by the abbreviation “PAH”

Thank you, this has been replaced.

2.Page 2 line 44 please development the abbreviations LIMK, TCTEX and SRC

Thank you, this has been now provided.

3.Page 2 line 46 could said BMPR2 loss instead of loss if BMPR2

Thank you, this has been changed.

4.Page 2 line 47 please replaces development of pulmonary hypertension by the abbreviation PAH

Thank you, this has been replaced.

5.Page 2 lines 55 please replace all of which is crucial to the development ….. by which are because “all it is plural not singular

Thank you, this has been replaced.

6.Page 2 line 62 add an abbreviation for nitric oxide

Thank you, this is now added.

7.Page 2 lines 63, 69 and 72, what is PH?

Thank you for pointing this out. This is now changed to pulmonary arterial hypertension(PAH).

8.Page 2 line 84 what is MYLK?

Thank you. MYLK is the myosin light chain kinase encoding gene. This is now added to the manuscript.

9.Page 2 lines 93 please add therefore the aim of this study was…..

Thank you. This is now changed.

10.Page 3, 4, 5, and 7 please add in each figure the corresponding abbreviations

I find it difficult to believe that the p of the significant differences is so exact throughout the entire manuscript, when one performs a statistical test, the p varies for this reason I think it is better to put them as p≤….. Please substitute this in all manuscript

We appreciate your comment and agree that there may be large variation between significance of 0.05 and 0.001. We have ensured that all statistical analysis meets our significance criteria for a p-value of less than 0.05 and none of the indicated values are equal to 0.05 which we would not consider significant so we have kept p<0.05, but as suggested we have substituted for the values that show greater significance using the less than or equal sign of p≤ 0.01 and p≤ 0.001.

I reviewed each of the westerns that were provided to me, but I never saw the p-MLC one in figure 1 and 4, please provide them, and the one provided in figure 4 does not correspond

We appreciate your careful review and realize that the p-MLC included in figure 1 was not provided in the original blots, we were able to identify the scanned and cropped blot that included all lanes of the experiment but the original full film cannot be located. As such we have removed that from Figure 1A and replaced with a new blot of p-MLC and corresponding actin that we think better represents are data as Figure 1B and reanalyzed the data to ensure accuracy of our data. We also included all 6 lanes for each of these experiments in order to compare 3 controls versus 3 experimental conditions to show some of the physiologic variability between experiments.

Regarding p-MLC blot for figure 4, after review the original blot was not well labeled on the original film and therefore we have added labels and identified p-MLC as protein identified near its molecular weight of 20 kDa.

11.Page 8 lines 292-299, please remove this paragraph and place it before starting the discussion because it can serve as an introduction to the discussion and replaces in same paragraph pulmonary arterial hypertension by PAH

Thank you, this is now placed at the beginning of the discussion.

12.Page 8, line 283 please substitute EC by VEC

Thank you for your comment, we have removed the acronym given that this is not used anywhere else in the manuscript.

13.Page 11, line 409 what sigma plot version is, please add

Thank you, SigmaPlot version 14 has been added as well as the company name.

14.Page 5 figure 3 (A) please delete this graphic and leave the b because it is not necessary, since it shows the same data

We appreciate your comment, but we feel that including the longitudinal data on normalized resistance by ECIS provides readers with additional information regarding rate (i.e. slope) of wound recovery and justifies the use of our time point at 12 hours to measure area under the curve, as such we have kept figure 3A in the current manuscript.

15.The load control (β-actin) in figure 4 and 5 are the same please substitute another load control in any of the figures

We appreciate your careful review. These figures are derived from the same experimental set and therefore have the same loading control, yet the findings have different purposes, to identify changes in cytoskeletal proteins and transcription factors in figure 4, followed by changes in proliferative markers in figure 5. As such, we would like to keep these figures separate for clarity to the reader. We have added language to the figure 5 legend to ensure to provide justification to the readers regarding the duplication of b-actin in this case.  

16.it is difficult to demonstrate so much significant value in figure 2 B with respect to the western of figure 2 A, because there is more load control (β-actin) in the lane that corresponds to the PCNA and there is more expression there, when making the quotient I think the result would not be as impressive, could you please review this result.

Thank you again for your careful review, all data were normalized to b-actin as loading control but we agree that there are small changes in b-actin, likely as a result of the hyper-proliferation of these cells, could impact normalization of results. We have now included the entire western blot with all six lanes in order to provide further transparency of our results. We re-analyzed the data with a second lab member and again found these to be statistically significant with a p-value of < 0.05, this updated bar graph is now included.

17.Please add a figure that summarize your manuscript for mayor comprehension

Thank you. This is now added as Figure 7 in the discussion section.

  1. I could add a part of the limitations of the study because a study of vascular reactivity in a chamber for an isolated organ would be great and would support the results obtained in cultured cells, I know in advance that this is difficult due to obtaining pulmonary arteries, but yes it was possible perhaps in postmortem patients

Thank you for your comments. We agree that there are limitations to using only cell models in the current manuscript. We utilized cell models that we think have strong relevance to human disease, particularly with reduction in BMPR2 levels as seen in numerous PAH patient subgroups as well as stimulation of the cells with a growth factor, VEGF, that is known to be elevated in patients with some forms of PAH. We have added additional information on these limitations for the project. Unfortunately we do not have access to post-mortem samples or explanted lungs at this time. We have added additional information in the discussion to clarify these limitations.

Round 2

Reviewer 2 Report

The authors have reviewed and responded to all comments appropriately. However, in the review process it appears that figures 4 and 6 overlapped please review and correct this.